biomechanics/bioengineering/biomedical engineering

plantar pressure, diabetic foot, pressure sensor, insole, buckling, honeycomb structure

**Author for correspondence:**
Panagiotis E. Chatzistergos
e-mail: panagiotis.chatzistergos@staffs.ac.uk

# A novel concept for low-cost non-electronic detection of overloading in the foot during activities of daily living

Panagiotis E. Chatzistergos and
Nachiappan Chockalingam

Centre for Biomechanics and Rehabilitation Technologies, Staffordshire University, Stoke-on-Trent, UK

PEC, 0000-0002-1580-0225; NC, 0000-0002-7072-1271

Identifying areas in the sole of the foot which are routinely overloaded during daily living is extremely important for the management of the diabetic foot. This work showcases the feasibility of reliably detecting overloading using a low-cost non-electronic technique. This technique uses thin-wall structures that change their properties differently when they are repeatedly loaded above or below a tuneable threshold. Flexible hexagonal thin-wall structures were produced using three-dimensional printing, and their mechanical behaviour was assessed before and after repetitive loading at different magnitudes. These structures had an elastic mechanical behaviour until a critical pressure ($P_{crit}$ = 252 kPa ± 17 kPa) beyond which they buckled. Assessing changes in stiffness after simulated use enabled the accurate detection of whether a sample was loaded above or below $P_{crit}$ (sensitivity = 100%, specificity = 100%), with the overloaded samples becoming significantly softer. No specific $P_{crit}$ value was targeted in this study. However, finite-element modelling showed that $P_{crit}$ can be easily raised or lowered, through simple geometrical modifications, to become aligned with established thresholds for overloading (e.g. 200 kPa) or to assess overloading thresholds on a patient-specific basis. Although further research is needed, the results of this study indicate that clinically relevant overloading could indeed be reliably detected without the use of complex electronic in-shoe sensors.

# 1. Background

People with diabetes tend to gradually lose the protective sensation of pain in their feet. As a result, they load their feet more heavily compared to their non-diabetic counterparts [1] and injured areas keep being loaded until the development of diabetic foot ulceration (DFU). DFU is an open wound which can have limited capacity for healing and it can get infected and lead even to amputation. Diabetes is the most common non-traumatic cause for amputation worldwide. In the UK alone, 169 people have a toe, foot or limb amputated as a result of diabetes every week, yet importantly up to 80% of these amputations could have been prevented with correct management [2,3].

Even though the development of DFU is multifactorial, the consensus is that its main mechanical trigger is increased plantar pressure. Owing to the key role repetitive overloading plays in the development of DFU, the reduction of plantar pressure is an important therapeutic objective in the clinical management of the diabetic foot.

To this end, therapeutic footwear and orthoses are commonly used to redistribute plantar loading and to protect critical areas of the foot from high pressures. While therapeutic footwear and orthoses have been used for some time now, their design is still mainly based on the clinical intuition of the prescribing clinician [4–7]. To promote evidence-based design of offloading interventions for the diabetic foot, thresholds of potentially injurious plantar pressure have been proposed in the literature [8–13]. Using plantar pressure measurements to detect and to offload areas of the foot that are subjected to pressures higher than these thresholds was shown to reduce the likelihood of recurrent DFU significantly more than offloading interventions that were not designed based on plantar pressure measurements [13]. The importance of plantar pressure for effective offloading is further reinforced by *in vivo* [14,15] and computational studies [16], indicating that the stiffness of the cushioning materials which are used in therapeutic footwear/orthoses can be optimized using information on plantar loading to maximize their capacity for offloading.

Despite mounting evidence in the literature and the publication of updated international guidelines recommending the use of plantar pressure measurements to inform footwear/orthoses prescription [11], such measurements are still not part of standard clinical practice. This is mainly owing to the complexity, cost and time associated with in-shoe plantar pressure measurements. Existing systems for measuring in-shoe pressure distribution use expensive electronic sensors which, in most cases, also mean that the patient has to be tethered to a data logging device. As a result, their use has been restricted within gait laboratories where the measured loads could be very different compared to those applied to the foot outside these controlled environments [6]. Indeed, in the vast majority of diabetic foot studies, where in-shoe pressure sensors were used, plantar pressure was only measured for a limited number of steps during walking in a straight line and on a level surface. Even though these measurements provide an accurate snapshot of plantar loading during testing, they might not be a representative assessment of the actual loading imposed on the soft tissue of the foot during a typical day in a person's life [6]. According to the literature, a measurement of accumulative loading during activities of daily living would be a more reliable indicator for the risk of DFU and therefore a better guide for clinical decision-making [6]. Sensor systems designed to provide continuous feedback to patients on plantar pressure could potentially be used to this end, but their cost and complexity is very likely to remain a barrier for widespread clinical use [17–21].

To address this challenge, the present paper proposes a novel concept for a low-cost pressure sensor that could be used to identify plantar areas which are routinely overloaded during activities of daily living. The proposed concept is based on the use of a sensor-insole comprising sensing elements that changes their properties when loaded above a predefined tuneable threshold. To stimulate further research in this direction, this work showcases the feasibility of producing such sensing elements using thin-wall structures and provides a discussion on the mechanical principles and limitations that underpin their function.

# 2. Methods

## 2.1. Non-electronic sensor-insole

Thin-wall structures exhibit distinctively different mechanical behaviour for different magnitudes of compressive loading. The hexagonal structures of figure 1 have a spring-like elastic response to loading for relatively low pressures but if pressure increases beyond a critical value ($P_{\text{crit}}$) they buckle

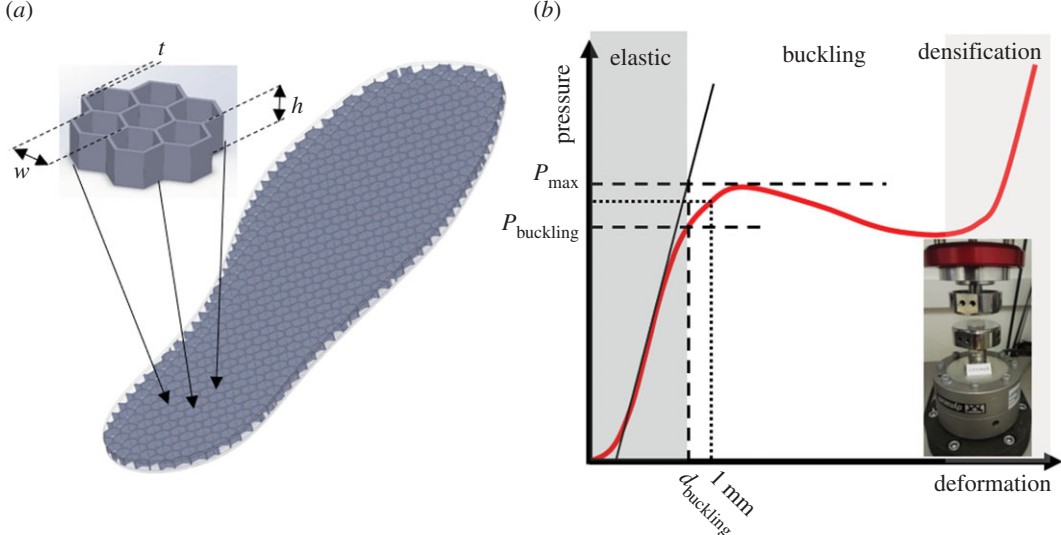

**Figure 1.** (*a*) The sensor-insole with its internal structure and key dimensions (*w*: hexagonal element width, *t*: wall thickness, *h*: insole thickness) made visible. The design of the samples comprising seven hexagonal elements which were used for mechanical testing is also shown (inset). (*b*) The mechanical response of the hexagonal thin-wall elements. The parameters that were used to describe the mechanical behaviour of the samples are noted on the graph. $P_{max}$: the maximum pressure before the total collapse of the thin-wall elements. $P_{buckling}$, $d_{buckling}$: the transition between elastic behaviour and buckling [22,23]. *K*: equivalent stiffness was assessed by the force that is needed to compress the insole by 1 mm.

and their resistance to loading drops, creating a local maximum in the pressure–deformation graph ($P_{max}$). Once this peak has been reached the structure becomes unstable and compression increases even without any increase in pressure. This instability is continuous until contact between walls and ultimately the bottoming-out of the structure gradually increase its stiffness again (i.e. densification).

When these thin-wall elements are subjected to repetitive loading they are likely to gradually accumulate microdamage which will eventually change their mechanical properties. Because stresses and strains will be significantly more intense in the elements that buckle compared to those that do not buckle, areas where pressure is routinely higher than $P_{crit}$ should accumulate damage at a significantly higher rate compared to areas where loading has been below that threshold.

The patient will be required to wear the sensor-insoles in their everyday footwear for a representative period (e.g. for a day or a week) before returning them for analysis. During the analysis of the sensor-insole, plantar areas that were routinely subjected to pressures higher than $P_{crit}$ should be identifiable, against those where pressure was below that threshold, by mapping changes in the mechanical properties of the sensor-insole.

## 2.2. The effect of exposure to loading

The capacity of hexagonal thin-wall structures to change their mechanical properties when subjected to repetitive loading in a way that would enable detecting overloading was tested in a series of mechanical tests.

Twenty-six samples comprising seven hexagonal elements each ($w = 6.5$ mm, $t = 0.5$ mm, $h = 5$ mm) were three-dimensionally printed from a thermoplastic polyurethane (TPU) (Filaflex, Recreus Industries S.L., filament Shore hardness: 72A) using fused deposition modelling (figure 1). A three-dimensional printable file of the samples used in these tests can be found in the electronic supplementary material SA. Their baseline mechanical behaviour was assessed by compressing them by 50% at a loading rate that simulates walking (200% sample thickness s⁻¹) [24]. To account for the effect of preconditioning, 10 load/unload cycles were performed and the final load cycle was used to draw the samples' pressure–deformation graph [25]. This graph was used to calculate their baseline $P_{max}$ and equivalent stiffness (*K*). *K* was assessed as the pressure that is needed to compress the sample by 1 mm. All samples were then mechanically aged and the effect of different exposures to loading on their mechanical behaviour was assessed. More specifically the first five samples were randomly assigned loading magnitudes of 50%, 75%, 100%, 125% and 150% of their respective $P_{max}$ and subjected to 2000

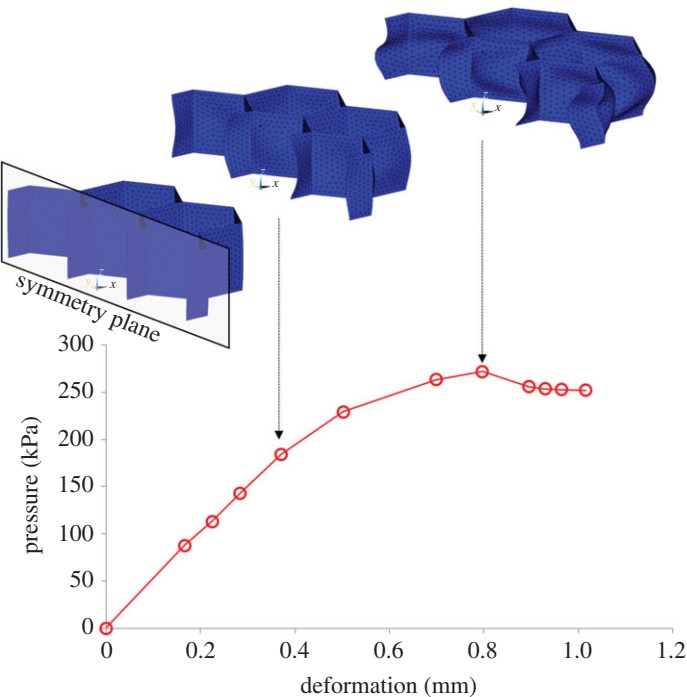

**Figure 2.** The numerically simulated pressure–deformation graph for a sample with the same geometry as the one used in mechanical testing ($w = 6.5$ mm, $t = 0.5$ mm, $h = 5$ mm). The FE model used in this simulation is also shown in its original/ undeformed shape and in two different deformed states.

and then to 12 000 more loading/unloading cycles (i.e. 14 000 in total) at 1 Hz to simulate a day's and a week's use by a person with diabetes [26]. The remaining 21 samples were randomly assigned different magnitudes of loading, ranging between 60% and 150% of the average $P_{max}$ and subjected to 2000 loading/unloading cycles [26]. After each mechanical ageing session, the samples were allowed to 'rest' for 24 h before measuring their mechanical behaviour again using the same testing procedure as at baseline.

## 2.3. Tuneability

The capacity to adapt the threshold for overloading by changing the geometry of the hexagonal elements was investigated in a finite-element (FE) analysis.

A three-dimensional FE model of the samples that were used in mechanical testing was created using shell elements. Taking advantage of the symmetry of geometry only half of the sample was meshed using 4670 elements (figure 2). A uniform element size was adopted based on a preliminary convergence analysis to ensure that the results are not dependent on mesh density. The mechanical behaviour of TPU was simulated as incompressible hyperelastic using the Ogden model (2nd order):

$$W = \sum_{i=1}^{2} \frac{\mu_i}{\alpha_i}(\bar{\lambda}_1^{\alpha_i} + \bar{\lambda}_2^{\alpha_i} + \bar{\lambda}_3^{\alpha_i} - 3),$$

where $W$ is the strain energy density, $\bar{\lambda}_p^a(p = 1,2, 3)$ are the deviatoric principal stretches and $\mu_i(i = 1,2)$ and $\alpha_i(i = 1,2)$ are the material coefficients defining the mechanical behaviour of the material. The material coefficients were assigned from the literature as follows: $\mu_1 = 0.133$ MPa, $\alpha_1 = 3.05$, $\mu_2 = -1214$ MPa and $\alpha_2 = -0.0054$ [27].

A preliminary investigation indicated that the simulation of contact with friction [28] between TPU and the compression plates had only a marginal effect on results and was therefore omitted from the analysis. Instead, the displacement of the model's nodes on the interface with the lower compression plate was fully constrained while those on the interface with the upper plate were allowed to move only in the direction of compression. A relatively large compressive displacement was imposed to ensure the model will be deformed to a point of instability as indicated by a clear peak in the pressure–deformation graph. To assess the effect of geometry on the function of the hexagonal structure (figure 1) simulations were run for different thicknesses of the thin walls

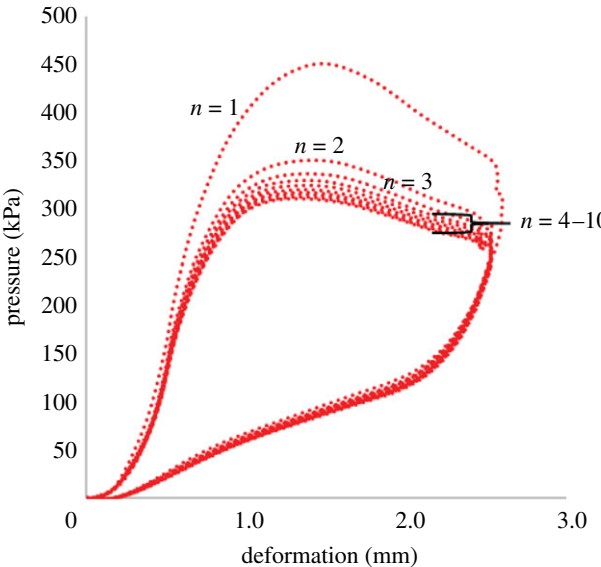

**Figure 3.** Typical pressure–deformation graph for load/unload cycles one ($n = 1$) to ten ($n = 10$).

(0.3 mm $\leq t \leq$ 0.6 mm, increments of 0.1 mm) and different hexagon widths (4.5 mm $\leq w \leq$ 7.5 mm, increments of 0.5 mm).

# 3. Results

## 3.1. The effect of exposure to loading

The samples were significantly stiffer during the first load/unload cycle, but their behaviour quickly stabilized (figure 3). Indicatively the $P_{max}$ for the first and last load/unload for the sample of figure 3 was 45% higher than the last load/unload cycle, but this difference dropped to 12% and to 8% for cycles two and three.

The results from the first five samples showed a trend of increasing change in $K$ with increasing loading magnitude for the samples which were loaded at 100%, 125% and 150% of their $P_{max}$. The properties of the samples which were loaded at 50% and 75% do not show significant changes in $K$. The difference between the two groups of samples is more pronounced following a week's simulated use but it also appears to be measurable after only a day's simulated use (figure 4). The raw data from mechanical testing before and after simulated use for these five samples can be found in the electronic supplementary material SB.

The average ($\pm$ s.d.) of baseline $P_{max}$ and $K$ of the remaining 21 samples was 304 kPa ($\pm$ 20 kPa) and 294 kPa ($\pm$ 19 kPa), respectively. Out of these 21 samples, 13 were overloaded (i.e. buckled) during simulated use of one day at random loading magnitudes. The overloaded samples exhibited a significantly higher change in their mechanical characteristics (figure 5$a$). When the change in $K$ is plotted over the sample-specific relative magnitude of loading during mechanical ageing ($P_{ageing}/P_{max}$), it appears that the step change in $K$ between samples which were overloaded and those that were not overloaded happens at a specific value of relative loading (figure 5$b$). Receiver operating characteristic curve analysis showed that the change in equivalent stiffness ($K$) can be used to detect whether a sample had been loaded above or below a threshold of $0.83^*P_{max}$ ($p < 0.001$, area = 1, sensitivity = 100%, specificity = 100%), which on average corresponded to a pressure threshold ($P_{crit}$) of 252 kPa ($\pm$ 17 kPa). The raw data from mechanical testing before and after simulated use for these 21 samples can be found in the electronic supplementary material SC.

## 3.2. Tuneability

The FE simulations produced pressure–deformation graphs with a clear peak which is similar to the experimental results (figure 2). $P_{max}$ increased linearly with the thickness of the thin walls ($t$) and decreased linearly with the width of the hexagons ($w$) (figure 6), indicating that these two geometrical

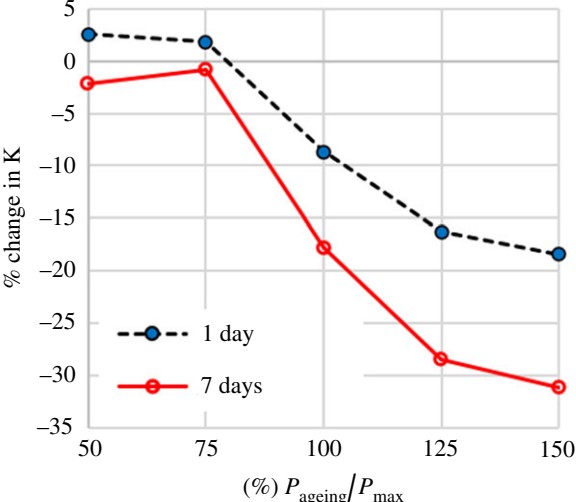

**Figure 4.** The effect of simulated use for 1 or 7 days on the equivalent stiffness ($K$) for different magnitudes of the applied ageing load. The magnitude of ageing load is presented as the % ratio of the maximum applied pressure in each load/unload cycle ($P_{ageing}$) over the maximum pressure of the sample's pressure–deformation graph ($P_{max}$).

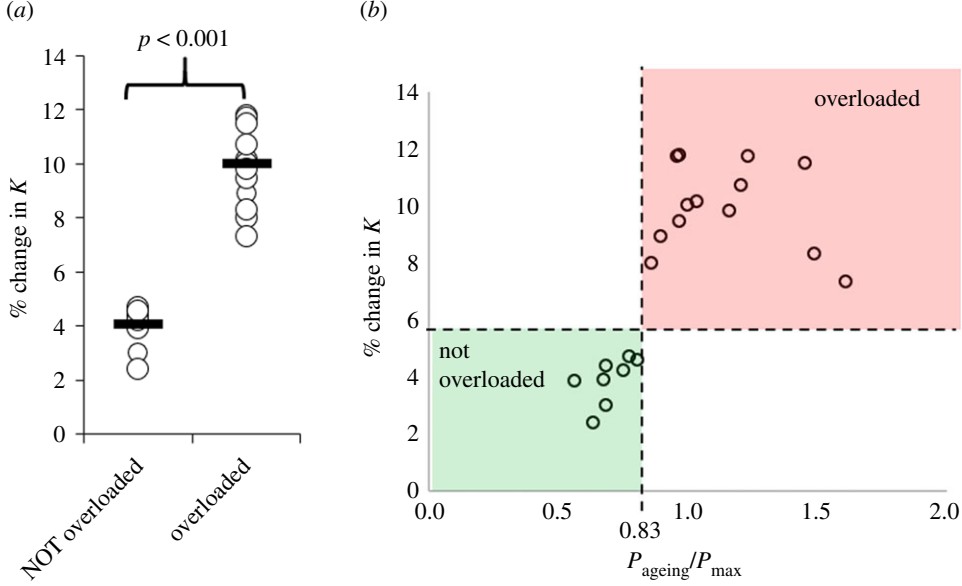

**Figure 5.** ($a$) Scatterplot highlighting the significantly higher change in equivalent stiffness ($K$) in the samples that were overloaded during simulated use [29]. Open circles show measurements for each sample and black lines show the group medians. ($b$) Scatterplot highlighting the increase in the change of $K$ when loading exceeds a threshold for overloading ($P_{crit} = 0.83^*P_{max}$). In this graph, $P_{max}$ corresponds to the sample-specific maximum pressure of the pressure-deformation graph.

parameters could be used to raise or lower the value of $P_{max}$ and with it also the value of $P_{crit}$. The effect of thickness on $P_{max}$ appears to be significantly stronger than that of element width. More specifically, 0.1 mm change in wall thickness led to 81 kPa change in $P_{max}$, highlighting the need for high manufacturing precision.

## 4. Discussion

The results of this study indicate that repeated overloading can be reliably detected by measuring changes in the mechanical behaviour of the hexagonal thin-wall structures presented here. More specifically, a total number of 2000 loading cycles, generated measurable differences between

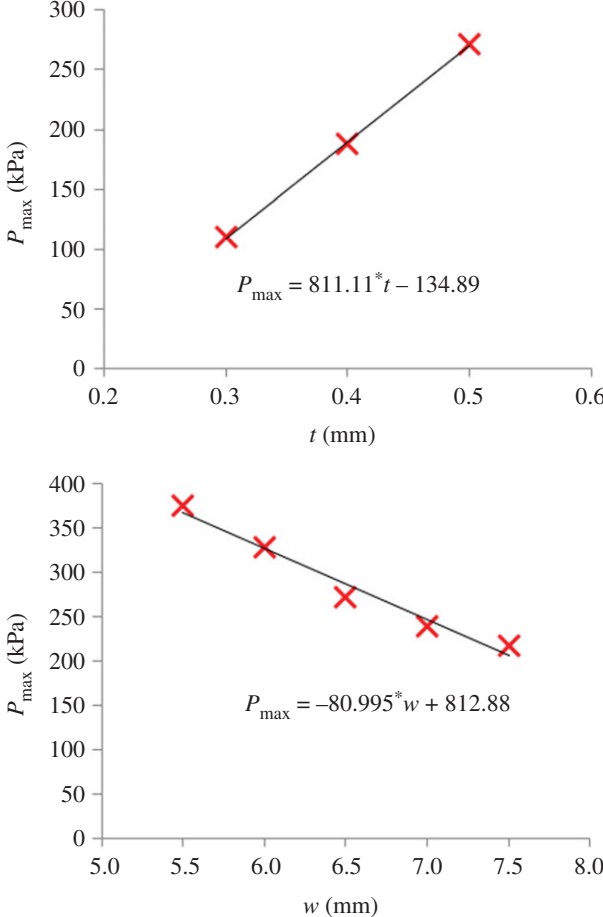

**Figure 6.** The effect of wall thickness ($t$) and of the width of the hexagonal elements ($w$) on the sample's $P_{max}$ value.

overloaded and not overloaded samples. The fact that the average daily step-count for a person with diabetes is also estimated to be 2000 steps [26] highlights the feasibility of using the presented sensor elements to assess overloading during a typical day in the life of a person with diabetes. Being able to get reliable results on overloading without the need for using the sensor for prolonged periods significantly enhances the clinical viability of the proposed sensing concept.

For the hexagonal structures tested here, the threshold for overloading ($P_{crit}$) was found to be equal to 252 kPa ($\pm$ 17 kPa). However, FE analysis demonstrated that this threshold can be easily raised or lowered through simple geometrical modifications depending on the requirements of the user. Even though no specific overloading threshold was targeted in this study, the measured $P_{crit}$ was relatively close to clinically relevant thresholds linked to ulceration risk and effective offloading of the at-risk foot. More specifically, current international guidelines on the prevention of DFU highlight pressure reduction below a threshold of 200 kPa as one of the criteria for effective offloading [11]. This recommendation is supported by two randomized control trials on the effectiveness of custom-made therapeutic footwear/orthoses [12,13]. At this point, it needs to be noted that, pressure reduction below 200 kPa is not the only criterion for effective offloading and more research is needed to establish pressure thresholds for increased risk for ulceration in the broader diabetic population [11]. The proposed sensing concept with its potential capacity for use in large cohorts outside the laboratory could contribute to this end.

Clearly, the value of $P_{crit}$ plays a key role in the proposed sensing concept. Therefore, a thorough exploration of its physical/engineering meaning is also warranted. The $P_{crit}$ of the hexagonal thin-wall structure tested here corresponded to 83% of the peak pressure ($P_{max}$) from the sample's pressure–deformation graph (figure 1). $P_{max}$ indicates the point in the sample's behaviour beyond which an increase in applied pressure is not needed for increased compression. When $P_{max}$ is reached the sample becomes unstable but buckling itself has started earlier than that. A method used in the literature to identify the onset of buckling in nonlinear materials uses the intersection between the linear part of the sample's response and a line parallel to the deformation axis at $P_{max}$ (figure 1) to find the strain or deformation at which buckling starts ($d_{buckling}$) [22,23]. When this method was followed, the value of

the pressure for the onset of buckling ($P_{buckling}$) was calculated to be, on average, equal to 82% ($\pm$ 14%) of $P_{max}$. Based on that it can be concluded that the calculated critical value of pressure beyond which the properties of the hexagonal structure change significantly corresponds to the pressure for the onset of buckling (i.e. $P_{crit} = P_{buckling}$).

In this study, the changes in the mechanical behaviour of samples were quantified by measuring changes in equivalent stiffness ($K$) using a load frame. Moving forward, a specialized automated device capable of mapping $K$ across the surface of a sensor-insole will be needed to eliminate the need for access to such complex and sophisticated infrastructure for general mechanical testing. Although other measurements could also be used (e.g. the slope of the linear part of the pressure–deformation graph), measuring pressure for a single deformation to assess $K$ significantly simplifies the entire assessment process which, in turn, can also reduce the complexity of any future specialized assessment device for the mapping of $K$.

FE analysis indicated that the threshold for detecting overloading can be raised or lowered according to the needs of a specific application, or even on a patient-specific basis, by changing the width of the hexagons or the thickness of their vertical walls. Out of these two parameters, wall thickness appeared to have the strongest effect on the mechanical behaviour of the sample. More specifically, it was found that changing wall thickness by as little as 0.1 mm can change $P_{max}$ by 81 kPa which translates in a change in $P_{crit}$ of 67 kPa. This finding highlights the need for robust quality control to avoid deviations between the indented and achieved $P_{crit}$.

Having thicker walls would make manufacturing easier, but at the same time, it could also have a detrimental effect on the function and clinical viability of the proposed sensor-insole. This is because the thickness of the walls ($t$) is linked to their height ($h$) which in turn needs to be kept as low as possible to enable use inside footwear. More specifically a ratio of $t/h \leq 0.1$ is needed to ensure the sensor elements will exhibit the buckling behaviour of a thin-wall structure.

The three-dimensional printed samples in this study had a wall thickness ($t$) of 0.5 mm for a height ($h$) of 5 mm. Considering that material inserts used in diabetic footwear have a thickness of up to 9 mm, a 5 mm tall sensor element could potentially be used inside a shoe. At the same time, having a relatively thick sensor-insole is likely to make the fitting of the sensor into everyday footwear challenging and affect the distribution of plantar pressure. A reduction in element height and therefore a reduction in wall thickness is needed to enhance the clinical viability of the proposed sensing concept.

In this study, three-dimensional printing was used because of its unique capabilities for prototyping. However further research will be needed to identify the optimum manufacturing process and materials which will enable reliably and sustainably producing even thinner walls. Appropriate selection of a recyclable material for the manufacturing of the sensor-insole can also minimize the potential environmental impact of the proposed sensor concept.

This study presented a proof of concept for the use of thin-wall structures to identify areas that are routinely loaded above a tuneable threshold. Even though only hexagonal elements were studied here, the findings are transferable to any thin-wall structure which has a well-defined threshold for buckling. Such structures could be potentially used in sensor-insoles, as a screening tool for overloading and to guide the prescription and design of custom-made therapeutic footwear/orthosis. The management of the foot-at-risk owing to diabetes appears to be one of the obvious applications. However, the proposed low-cost sensor concept could also be applicable in other clinical or ergonomic applications where understanding of plantar loading during activities of daily living is needed (e.g. industry-specific footwear).

In this study, the effect of use (for a day or a week) was simulated by repetitive cycles of compressive loading at room temperature. Further research will be needed to assess the effect of the in-shoe environment (i.e. temperature and humidity) and loading conditions including the effect of shear loading.

# 5. Conclusion

Going against tradition this study suggests an alternative approach to plantar pressure measurements for clinical applications that do not rely on expensive electronic equipment. Even though more research is still needed towards a clinically applicable sensor solution, the results presented here indicate that the buckling of thin-wall structures can be used to produce sensor elements which change their mechanical properties when repeatedly loaded above a pressure threshold. This threshold corresponds to the pressure that is needed to cause buckling and can be easily tuned to the needs of a specific application or even on a patient-specific basis by adapting the geometry of the structure. Subject to

further research for the reliable and sustainable manufacturing of thin sensor elements that do not affect the distribution of plantar loading and the automation of its analysis process; the proposed sensor concept could be used to identify areas that are routinely overloaded during activities of daily living to enhance ulceration risk assessment and inform the design of effective offloading interventions.

Data accessibility. Data supporting this research are provided as the electronic supplementary material. The three-dimensional printable file (*.stl) that was used to produce the samples is included in the electronic supplementary material SA. The mechanical testing results for the first five samples (samples A to E) which were subjected to 1 and 7 days of simulated use can be found in the electronic supplementary material SB.xlsx. The mechanical testing results for the next 21 samples (samples 1 to 21) which were subjected to 1 day of simulated use at random magnitudes of loading can be found in the electronic supplementary material SC.xlsx. In both *.xlsx files, the information about the loading magnitude that was applied to each sample can be found in a separate tab (labelled: samples).

Authors' contributions. P.E.C. conceived the study, designed the study, coordinated data collection and drafted the manuscript. N.C. participated in the design of the study and data collection/analysis and helped draft the manuscript. Both authors gave final approval for publication.

Competing interests. Authors (P.E.C. and N.C.) are named co-inventors in a relevant patent (Publication no. WO/2019/073261).

Funding. This work was supported by Innovate UK (grant no. 16001], a project in collaboration with Cadscan Limited, Chester, UK.

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
