## [Peer Review File · Royal Society Open Science]

Review History

RSOS-202035.R0 (Original submission)

Review form: Reviewer 1 (Isabel Sacco)

Is the manuscript scientifically sound in its present form?

Yes

Are the interpretations and conclusions justified by the results?

Yes

Is the language acceptable?

Yes

Do you have any ethical concerns with this paper?

No

Have you any concerns about statistical analyses in this paper?

No

Recommendation?

Accept with minor revision (please list in comments)

Comments to the Author(s)

General Comments

The goal of this manuscript was to showcase the feasibility of reliably detecting overloading using a low-cost non-electronic technique that uses thinwall structures that change their properties when they are routinely loaded above or below a tuneable threshold. The manuscript is solidly grounded and the proposal measurement technique is relevant. This is a bold and audacious proposal that will break paradigms on how to assess the risk of ulcers in people with diabetes and, perhaps, even change the modus operandi of regular evaluations needed for these people to avoid further complications such as amputations resulting from worsening of the disease.

Although the introduction discussion need some additional work, the paper is straightforward and could be improved. There are some suggestions/comments that I addressed throughout the manuscript. Please, consider reviewing them.

Abstract

1. In the abstract, it is important to describe clearly the threshold (Pcrit) adopted (or how they can be interchanged) and the rationale for it.
2. Please, add the value for buckling which was surprisingly (and fortunately) very close to the suggested threshold for ulcer risk in the international guidelines. Even if this threshold is still questionable, it would be important to mention that in your paper: abstract and conclusion, as it makes your technique more plausible and considered adoptable worldwide.

Introduction

1. "Due to the key role repetitive overloading plays in the development of DFU, the reduction of plantar pressure is an important therapeutic objective in the clinical management of diabetic foot." Please, cite references at the end of this sentence to support your statement. I would suggest the more recent systematic review Netten, Jaap J.; Sacco, Isabel C.N. ; Lavery, Lawrence A.; Monteiro'soares, Matilde ; Rasmussen, Anne ; Raspovic, Anita ; Bus, Sicco A. Treatment of modifiable risk factors for foot ulceration in persons with diabetes: a systematic review. *Diabetes-Metabolism Research And Reviews*, v. 37, p. e3271, 2020.
2. "To promote evidence-based design of offloading interventions for the diabetic foot, thresholds of potentially injurious plantar pressure have been proposed in the literature (7-9)." Please, consider adding an important and recent reference from the guidelines that explicitly suggest a threshold, although still subject to criticism: Bus, Sicco A.; Lavery, Lawrence A. ; Monteiro'soares, Matilde ; Rasmussen, Anne; Raspovic, Anita; Sacco, Isabel C.N. ; Netten, Jaap J. Guidelines on the prevention of foot ulcers in persons with diabetes (IWGDF 2019 update). *Diabetes-Metabolism Research And Reviews*, v. 36, p. e3269, 2020.
3. "Despite the mounting evidence in support of the use of plantar pressure measurements to inform footwear/orthoses prescription, such measurements are still not part of standard clinical practice." Consider adding a statement like "..., despite it is also recommended in the updated International Guidelines...

Sensor development session

1. Page 7, Line 10 - Please be consistent with the use of Pcrit (used in the text, abstract) or Pcr.
2. Consider adding a paragraph discussing about the time needed to accumulate microdamage that would change the mechanical properties of the material, because if this is too much, maybe a DFU could occur before the material changes it mechanical linear properties, and

the main aim of the insole to detect high pressures before an injury occur, would be compromised.

3. The authors should consider explaining in this session (or in the methods) the routine imposed by the use of this new sensor to detect high pressures, that includes the patient to go to a specialized lab (not any lab) to evaluate the physical properties of the sensor after one day of receiving/using the sensor.

Methods

1. The first paragraph is repeated in the previous session (1st paragraph of page 7). Maybe keeping only in the previous session would be enough or, even better, try to group the session 2 with session 3 (methods) and explain the sensor development in one only session.

Discussion and conclusion

1. Overall, the discussion is straightforward and focused. My only suggestion is to discuss the values obtained in the tests with the suggested threshold in the literature, particularly in the International Guidelines. The value for buckling was surprisingly (and fortunately) very close to the suggested threshold for ulcer risk in the international guidelines. Even if this threshold is still questionable, it would be important to mention that in your paper: discussion and conclusion, as it makes your technique more plausible and considered adoptable worldwide.

2. As one potential limitations of the technique, it would be interesting to add the need for the patient go to a very specialized lab that could measure the properties of the insole after the first day usage.

Decision letter (RSOS-202035.R0)

Dear Dr Chatzistergos

The Editors assigned to your paper RSOS-202035 "A novel concept for low-cost non-electronic detection of overloading in the foot during activities of daily living." have now received comments from reviewers and would like you to revise the paper in accordance with the reviewer comments and any comments from the Editors. Please note this decision does not guarantee eventual acceptance.

Please submit your revised manuscript and required files (see below) no later than 21 days from today's (ie 26-Mar-2021) date. Note: the ScholarOne system will 'lock' if submission of the revision is attempted 21 or more days after the deadline. If you do not think you will be able to meet this deadline please contact the editorial office immediately.

on behalf of Dr Maria Charalambides (Associate Editor) and R. Kerry Rowe (Subject Editor)
openscience@royalsociety.org

Reviewer comments to Author:

Reviewer: 1
Comments to the Author(s)

General Comments

The goal of this manuscript was to showcase the feasibility of reliably detecting overloading using a low-cost non-electronic technique that uses thinwall structures that change their properties when they are routinely loaded above or below a tuneable threshold. The manuscript is solidly grounded and the proposal measurement technique is relevant. This is a bold and audacious proposal that will break paradigms on how to assess the risk of ulcers in people with diabetes and, perhaps, even change the modus operandi of regular evaluations needed for these people to avoid further complications such as amputations resulting from worsening of the disease.

Although the introduction discussion need some additional work, the paper is straightforward and could be improved. There are some suggestions/comments that I addressed throughout the manuscript. Please, consider reviewing them.

Abstract

1. In the abstract, it is important to describe clearly the threshold (P_{crit}) adopted (or how they can be interchanged) and the rationale for it.
2. Please, add the value for buckling which was surprisingly (and fortunately) very close to the suggested threshold for ulcer risk in the international guidelines. Even if this threshold is still questionable, it would be important to mention that in your paper: abstract and conclusion, as it makes your technique more plausible and considered adoptable worldwide.

Introduction

1. "Due to the key role repetitive overloading plays in the development of DFU, the reduction of plantar pressure is an important therapeutic objective in the clinical management of diabetic foot." Please, cite references at the end of this sentence to support your statement. I would suggest the more recent systematic review Netten, Jaap J.; Sacco, Isabel C.N. ; Lavery, Lawrence A.; Monteiro'soares, Matilde ; Rasmussen, Anne ; Raspovic, Anita ; Bus, Siccio A. Treatment of

modifiable risk factors for foot ulceration in persons with diabetes: a systematic review. *Diabetes-Metabolism Research And Reviews*, v. 37, p. e3271, 2020.

2. "To promote evidence-based design of offloading interventions for the diabetic foot, thresholds of potentially injurious plantar pressure have been proposed in the literature (7-9)." Please, consider adding an important and recent reference from the guidelines that explicitly suggest a threshold, although still subject to criticism: Bus, Sicco A.; Lavery, Lawrence A. ; Monteiro'soaes, Matilde ; Rasmussen, Anne; Raspovic, Anita; Sacco, Isabel C.N. ; Netten, Jaap J. Guidelines on the prevention of foot ulcers in persons with diabetes (IWGDF 2019 update). *Diabetes-Metabolism Research And Reviews*, v. 36, p. e3269, 2020.

3. "Despite the mounting evidence in support of the use of plantar pressure measurements to inform footwear/orthoses prescription, such measurements are still not part of standard clinical practice." Consider adding a statement like "..., despite it is also recommended in the updated International Guidelines...

Sensor development session

1. Page 7, Line 10 - Please be consistent with the use of Pcrit (used in the text, abstract) or Pcr.

2. Consider adding a paragraph discussing about the time needed to accumulate microdamage that would change the mechanical properties of the material, because if this is too much, maybe a DFU could occur before the material changes its mechanical linear properties, and the main aim of the insole to detect high pressures before an injury occurs, would be compromised.

3. The authors should consider explaining in this session (or in the methods) the routine imposed by the use of this new sensor to detect high pressures, that includes the patient to go to a specialized lab (not any lab) to evaluate the physical properties of the sensor after one day of receiving/using the sensor.

Methods

1. The first paragraph is repeated in the previous session (1st paragraph of page 7). Maybe keeping only in the previous session would be enough or, even better, try to group the session 2 with session 3 (methods) and explain the sensor development in one only session.

Discussion and conclusion

1. Overall, the discussion is straightforward and focused. My only suggestion is to discuss the values obtained in the tests with the suggested threshold in the literature, particularly in the International Guidelines. The value for buckling was surprisingly (and fortunately) very close to the suggested threshold for ulcer risk in the international guidelines. Even if this threshold is still questionable, it would be important to mention that in your paper: discussion and conclusion, as it makes your technique more plausible and considered adoptable worldwide.

2. As one potential limitation of the technique, it would be interesting to add the need for the patient to go to a very specialized lab that could measure the properties of the insole after the first day usage.

===PREPARING YOUR MANUSCRIPT===

===PREPARING YOUR REVISION IN SCHOLARONE===

- If you are providing image files for potential cover images, please upload these at this step, and inform the editorial office you have done so. You must hold the copyright to any image provided.
- A copy of your point-by-point response to referees and Editors. This will expedite the preparation of your proof.

- Ensure that your data access statement meets the requirements at <https://royalsociety.org/journals/authors/author-guidelines/#data>. You should ensure that you cite the dataset in your reference list. If you have deposited data etc in the Dryad repository, please include both the 'For publication' link and 'For review' link at this stage.
- If you are requesting an article processing charge waiver, you must select the relevant waiver option (if requesting a discretionary waiver, the form should have been uploaded at Step 3 'File upload' above).
- If you have uploaded ESM files, please ensure you follow the guidance at <https://royalsociety.org/journals/authors/author-guidelines/#supplementary-material> to include a suitable title and informative caption. An example of appropriate titling and captioning may be found at https://figshare.com/articles/Table_S2_from_Is_there_a_trade-off_between_peak_performance_and_performance_breadth_across_temperatures_for_aerobic_scope_in_teleost_fishes_/3843624.

Author's Response to Decision Letter for (RSOS-202035.R0)

See Appendix A.

RSOS-202035.R1 (Revision)

Review form: Reviewer 1 (Isabel Sacco)

Is the manuscript scientifically sound in its present form?

Yes

Are the interpretations and conclusions justified by the results?

Yes

Is the language acceptable?

Yes

Do you have any ethical concerns with this paper?

No

Have you any concerns about statistical analyses in this paper?

No

Recommendation?

Accept as is

Comments to the Author(s)

The paper sounds much better, the additions implemented contributed for improving the clarity and clinical usefulness of the results.

Decision letter (RSOS-202035.R1)

Dear Dr Chatzistergos,

It is a pleasure to accept your manuscript entitled "A novel concept for low-cost non-electronic detection of overloading in the foot during activities of daily living." in its current form for publication in Royal Society Open Science. The comments of the reviewer(s) who reviewed your manuscript are included at the foot of this letter.

on behalf of R. Kerry Rowe (Subject Editor)
openscience@royalsociety.org

Associate Editor Comments to Author:

Comments to the Author:

Congratulations on the acceptance of this manuscript - the reviewer is now satisfied that your manuscript is ready for publication. The journal's production team will be in touch shortly.

Reviewer comments to Author:

Reviewer: 1

Comments to the Author(s)

The paper sounds much better, the additions implemented contributed for improving the clarity and clinical usefulness of the results.

Appendix A

General Comments

The goal of this manuscript was to showcase the feasibility of reliably detecting overloading using a low-cost non-electronic technique that uses thinwall structures that change their properties when they are routinely loaded above or below a tuneable threshold. The manuscript is solidly grounded and the proposal measurement technique is relevant. This is a bold and audacious proposal that will break paradigms on how to assess the risk of ulcers in people with diabetes and, perhaps, even change the modus operandi of regular evaluations needed for these people to avoid further complications such as amputations resulting from worsening of the disease.

Although the introduction discussion need some additional work, the paper is straightforward and could be improved. There are some suggestions/comments that I addressed throughout the manuscript. Please, consider reviewing them.

The authors would like to thank the reviewer for their positive feedback, helpful comments and valuable suggestions. The manuscript is now revised to address the reviewer's comments. Each comment is followed by the response of the authors (in underlined italics) and a short description of the action undertaken. In case a part of the text is rewritten, or new material is added the new version is also included (centrally justified italics). For your convenience all revisions are highlighted in the revised manuscript.

Abstract

1. In the abstract, it is important to describe clearly the threshold (P_{crit}) adopted (or how they can be interchanged) and the rationale for it.
2. Please, add the value for buckling which was surprisingly (and fortunately) very close to the suggested threshold for ulcer risk in the international guidelines. Even if this threshold is still questionable, it would be important to mention that in your paper: abstract and conclusion, as it makes your technique more plausible and considered adoptable worldwide.

We agree with the reviewer's recommendations regarding the abstract. To address these two comments the value of the overloading threshold (P_{crit}) is added in the revised abstract. Moreover, the closing section of the abstract is also rewritten as follows to include a brief discussion into the relevance of P_{crit} with existing pressure thresholds used in diabetic foot management, as well as an explanation on how P_{crit} can be tuned and the rationale behind its tuning.

“No specific value of P_{crit} was targeted here. However finite element modelling showed that P_{crit} can be easily raised or lowered, through simple geometrical modifications, to become aligned with established thresholds for overloading (e.g. 200kPa) or to assess overloading thresholds on a patient-specific basis. Although further research is needed, the results of this study indicate that clinically relevant overloading could indeed be reliably detected without the use of complex electronic in-shoe sensors.”

Introduction

1. **“Due to the key role repetitive overloading plays in the development of DFU, the reduction of plantar pressure is an important therapeutic objective in the clinical management of diabetic foot.”** Please, cite references at the end of this sentence to support your statement. I would suggest the more recent systematic review Netten, Jaap J.; Sacco, Isabel C.N. ; Lavery, Lawrence A.; Monteiro’soares, Matilde ; Rasmussen, Anne ; Raspovic, Anita ; Bus, Sicco A. Treatment of modifiable risk factors for foot ulceration in persons with diabetes: a systematic review. *Diabetes-Metabolism Research And Reviews*, v. 37, p. e3271, 2020.

The recommended citation is now added to the revised manuscript.

2. **“To promote evidence-based design of offloading interventions for the diabetic foot, thresholds of potentially injurious plantar pressure have been proposed in the literature (7–9).”** Please, consider adding an important and recent reference from the guidelines that explicitly suggest a threshold, although still subject to criticism: Bus, Sicco A.; Lavery, Lawrence A. ; Monteiro'soares, Matilde ; Rasmussen, Anne; Raspovic, Anita; Sacco, Isabel C.N. ; Netten, Jaap J. Guidelines on the prevention of foot ulcers in persons with diabetes (IWGDF 2019 update). *Diabetes-Metabolism Research And Reviews*, v. 36, p. e3269, 2020.

The recommended citation is now added to the revised manuscript.

3. **“Despite the mounting evidence in support of the use of plantar pressure measurements to inform footwear/orthoses prescription, such measurements are still not part of standard clinical practice.”** Consider adding a statement like “..., despite it is also recommended in the updated International Guidelines...”

The aforementioned sentences are now rewritten as follows:

“Despite mounting evidence in the literature and the publication of updated international guidelines recommending the use of plantar pressure measurements to inform footwear/orthoses prescription (11), such measurements are still not part of standard clinical practice.”

Sensor development session

1. Page 7, Line 10 – Please be consistent with the use of P_{crit} (used in the text, abstract) or P_{cr} .

We would like to thank the reviewer for pointing out this inconsistency. The overloading threshold of the thin-wall structures is now referred to as P_{crit} across the manuscript.

2. Consider adding a paragraph discussing about the time needed to accumulate microdamage that would change the mechanical properties of the material, because if this is too much, maybe a DFU could occur before the material changes its mechanical linear properties, and the main aim of the insole is to detect high pressures before an injury occurs, would be compromised.

The reviewer is correct to point out the significance of time for the clinical viability of the proposed sensing concept. This is now clarified in the opening paragraph of “discussion” as follows:

“The results of this study indicate that repeated overloading can be reliably detected by measuring changes in the mechanical behaviour of the hexagonal thin-wall structures presented here. More specifically, a total number of 2,000 loading cycles, generated measurable differences between overloaded and not overloaded samples. The fact that the average daily step-count for a person with diabetes is also estimated to be 2,000 steps (26), highlights the feasibility of using the presented sensor elements to assess overloading during a typical day in the life of a person with diabetes. Being able to get reliable results on overloading without the need for using the sensor for prolonged periods significantly enhances the clinical viability of the proposed sensing concept.”

3. The authors should consider explaining in this session (or in the methods) the routine imposed by the use of this new sensor to detect high pressures, that includes the patient to go to a specialized lab (not any lab) to evaluate the physical properties of the sensor after one day of receiving/using the sensor.

In this study, changes in mechanical behaviour were measured using a load frame. The reviewer is correct to point out that the use of such infrastructure for general mechanical testing requires specialised expertise that can only be found in specialised laboratories. Moving forward, however, we believe that automating the mapping of changes in K is feasible. This will require the development and validation of a new specialised device in a separate developmental step towards a clinically applicable sensor system. At the same time, it is also

important to highlight that the fact that reliable assessment of overloading can be achieved with a single measurement of pressure for a single value of compression substantially improves the feasibility of such an approach. To clarify these points the following statements are now added in “discussion”:

“In this study, the changes in the mechanical behaviour of samples were quantified by measuring changes in equivalent stiffness (K) using a load frame. Moving forward, a specialised automated device capable of mapping K across the surface of a sensor-insole will be needed to eliminate the need for access to such complex and sophisticated infrastructure for general mechanical testing. Although other measurements could also be used (e.g. the slope of the linear part of the graph), measuring pressure for a single deformation to assess K significantly simplifies the entire assessment process which, in turn, can also reduce the complexity of any future specialised assessment device for the mapping of K.”

Methods

1. The first paragraph is repeated in the previous session (1st paragraph of page 7). Maybe keeping only in the previous session would be enough or, even better, try to group the session 2 with session 3 (methods) and explain the sensor development in one only session.

Sections 2 and 3 are now grouped as recommended.

Discussion and conclusion

1. Overall, the discussion is straightforward and focused. My only suggestion is to discuss the values obtained in the tests with the suggested threshold in the literature, particularly in the International Guidelines. The value for buckling was surprisingly (and fortunately) very close to the suggested threshold for ulcer risk in the international guidelines. Even if this threshold is still questionable, it would be important to mention that in your paper: discussion and conclusion, as it makes your technique more plausible and considered adoptable worldwide.

We would like to thank the reviewer for this very helpful suggestion. The following paragraph is now added in “discussion” to highlight the relevance of P_{crit} with existing thresholds used in the management of diabetic foot ulceration:

“For the hexagonal structures tested here, the threshold for overloading (P_{crit}) was found to be equal to 252kPa(± 17 kPa). However, FE analysis demonstrated that this threshold can be

easily raised or lowered through simple geometrical modifications depending on the requirements of the user. Even though no specific overloading threshold was targeted in this study, the measured Pcrit was relatively close to clinically relevant thresholds linked to ulceration risk and effective offloading of the at-risk foot. More specifically, current international guidelines on the prevention of DFU highlight pressure reduction below a threshold of 200kPa as one of the criteria for effective offloading (11). This recommendation is supported by two randomised control trials on the effectiveness of custom-made therapeutic footwear/ orthoses (12,13). At this point it needs to be noted that, pressure reduction below 200kPa is not the only criterion for effective offloading and more research is needed to establish pressure thresholds for increased risk for ulceration in the broader diabetic population (11). The proposed sensing concept with its potential capacity for use in large cohorts outside the laboratory could contribute to this end.”

2. As one potential limitations of the technique, it would be interesting to add the need for the patient go to a very specialized lab that could measure the properties of the insole after the first day usage.

This limitation is now acknowledged in the revised manuscript. More specifically a statement is now added in “conclusions” highlighting the need to automate the analysis process. A more extensive discussion on this point is also added in “discussion” (please also see our response to comment 3.Sensor development section)